# Blue Cheeses: Microbiology and Its Role in the Sensory Characteristics

Teresa María López-Díaz [1,2,*], Ángel Alegría [1], Jose María Rodríguez-Calleja [1,2], Patricia Combarros-Fuertes [1,2], José María Fresno [1,2], Jesús A. Santos [1,2], Ana Belén Flórez [3,4] and Baltasar Mayo [3,4]

1   Food Hygiene and Food Technology Department, Veterinary Faculty, University of León, 24071 León, Spain; a.alegria@unileon.es (Á.A.); jm.rcalleja@unileon.es (J.M.R.-C.); pcomf@unileon.es (P.C.-F.); jmfreb@unileon.es (J.M.F.); j.santos@unileon.es (J.A.S.)
2   Institute of Food Science and Technology, University of León, 24071 León, Spain
3   Department of Microbiology and Biochemistry, Dairy Research Institute of Asturias, Spanish National Research Council–IPLA-CSIC, 33300 Villaviciosa, Spain; abflorez@ipla.csic.es (A.B.F.); baltasar.mayo@ipla.csic.es (B.M.)
4   Institute of Sanitarian Research of Asturian Principality—ISPA, 33011 Oviedo, Spain
*   Correspondence: teresa.lopez@unileon.es

**Abstract:** Blue cheeses are those whose matrix is veined with a blue, blue-grey, or blue-green colour due to the development of *Penicillium roqueforti*. There are more than 45 varieties of blue cheese produced worldwide, with some distinct features, although the manufacture process is similar. In addition to *P. roqueforti*, complex microbial populations interact and succeed throughout the manufacturing and ripening at the cheese's surface (the rind) and interior (matrix). The microbiota of blue cheeses is made up of a vast array of both prokaryotic and eukaryotic microorganisms. Acidification of the curd relies on the action of lactococci and other lactic acid bacteria (LAB) species. The ripened cheeses' final quality and shelf-life properties largely depend on the enzymatic systems of the components of the microbiota, particularly on those of LAB, *P. roqueforti*, and yeast species. Proteolysis is the most complex and important primary biochemical process involved in blue-veined cheeses during ripening, with *P. roqueforti* being considered the main proteolytic agent. Lipolysis is also strong, originating, among other compounds, ketones, which are the main aroma compounds in blue-veined cheeses. In addition, several bioactive compounds are produced during ripening. The biochemical activities, mainly of microbial origin, are responsible for the sensory characteristics of these very appreciated cheese varieties worldwide.

**Keywords:** blue cheeses; *Penicillium roqueforti*; mycotoxins; proteolysis; lipolysis; cheese microbiota

## 1. Introduction

Blue cheeses (blue-veined cheeses) are characterized by the presence of blue veins in the interior due to the development of the fungus *Penicillium roqueforti* (naturally present or added as secondary culture). The first blue cheese described was Gorgonzola (Italy, ninth century), followed by Roquefort (eleventh century), although there are some reports that place the French cheese in the eight century [1]. The other varieties were first described from the 17th century onwards [2].

Blue cheeses are semi-hard cheeses, with a very variable weight depending on the type (from 0.3 to more than 10 kg), a fresh dry matter of 50–60%, a fat content of 30–40%, a protein content of 20–30%, and a variable NaCl content (most commonly, 3–4%) [1].

Manufacture of blue cheese has as differential points the following: addition of *P. roqueforti* spores (optional); addition of hetero-fermentative lactic cultures (*Leuconostoc* spp., optional, see Section 3.3); cutting of the curd into small pieces (to create an open paste); dry or, less frequently, brine-salting; piercing (allows air to enter, which activates the *P. roqueforti* growth and the blue veins development; optional) and ripening at around 10 °C, and 85–95% humidity (for some varieties, in natural caves), for at least 1–2 months [1–4].

## 2. Types of Blue Cheese

There are more than 45 varieties of blue cheese produced worldwide (https://en.wikipedia.org/wiki/List_of_blue_cheeses, accessed on 1 May 2023). Of them, the best-known are made in Europe (Roquefort, Cabrales, Stilton, Gorgonzola, and Danablu), all of which have been granted with Protected Designation of Origin (PDO) or Protected Geographical Indication (PGI) (Table 1). Apart from these, Spain produces other varieties such as Valdeon (PGI), Picón-Bejes Tresviso (PDO), and Gamonedo (PDO), all of them in the Picos de Europa mountain range (in the North) and France, the Bleu d'Auvergne (PDO), or the Bleu de Bresse, among 14 other varieties.

Blue cheeses may be made with raw or pasteurised milk (on some occasions, thermised Danablu) coming from cow, ewe or, occasionally, goat, or a mixture of them, although worldwide, cow's milk is most usual. For instance, in France most blue cheeses are made with cow's milk, except Roquefort, and in Spain and Greece, it is usual to add sheep´s milk to the cow´s milk. As for the use of raw milk, in some varieties it is mandatory according to the PDO (i.e., Cabrales or Roquefort) (Table 1) [1–4].

## 3. Microbiology

Blue cheese microbiota is complex, particularly when raw milk is used. Lactic acid bacteria (LAB) and fungi dominate the process. Among the first, apart from *Lactococcus* spp., *Leuconostoc* is usually present, which favors an open texture. Among the fungi, apart from *P. roqueforti*, different yeasts are usually present. All of them contribute to the characteristics of the final product.

### 3.1. Ecological Factors

The ecological factors that influence the microbiota of cheese are variable to some degree depending on the blue cheese variety. The pH increases during ripening, from 4.7–5.0 (first days of manufacture) to 6.0–7.0 at the end of ripening (the pH in the interior rises more rapidly than on the surface). This increase is due to the degradation of lactic acid by non-LAB moulds (*P. roqueforti*) and yeasts and proteolysis. Water activity decreases rapidly during the first week and slowly in the rest of the manufacture, ending in 0.91–0.94. As for the sodium chloride content, the range in the final product is 2–5% (most commonly, 3–4%) [1,2]. pH and salt in humidity gradients and low temperature are the physicochemical coordinates driving the development of secondary microbiotas and guiding the enzyme activities required for proper maturation.

### 3.2. Blue Cheese Microbiota

Blue cheese can still be manufactured in different sizes and shapes using milk from different mammal species (or mixtures) and following different manufacturing and ripening technologies [5]. The type of milk and the technological processing greatly influence the cheese microbiota during the manufacturing and ripening stages, and thus its final sensory attributes. Most traditional blue cheeses (e.g., Roquefort, Gorgonzola, Cabrales, Gamonedo, etc.) are still manufactured with raw milk following traditional artisan technologies. Therefore, the large diversity in microbial populations coupled with different aroma and taste profiles is not surprising. For standardization, strains of mesophilic LAB species (*Lactococcus lactis*, *Lc. cremoris*, and *Leuconostoc* spp.) are currently being added as starters for either raw milk-made and pasteurized milk-made cheeses [1]. As an exception, Gorgonzola starters are mixes of mesophilic (as above) and thermophilic (*Streptococcus thermophilus* and *Lactobacillus delbrueckii*) LAB species [6]. Traditionally, cheeses were not inoculated either with *P. roqueforti* spores. They became naturally contaminated with this fungus from manufacturing and ripening environments. However, adding commercial spores is common practice at both artisan and industrial manufacturing scales (Table 1). Other factors affecting the microbiota are the salting of the cheeses by applying coarse salt onto the surface or by milling and mixing the curd with salt prior to moulding [1], or brine immersion, as well as the ripening of blue cheese at low temperatures (8–12 °C) and high

relative humidity (>90%). Finally, cheeses are frequently pierced to facilitate the entry of air to allow a uniform development of *P. roqueforti* into the cheese matrix, which contributes to the typical visual aspect at cutting.

**Table 1.** Main properties of European, traditional blue-veined cheeses and major microbial populations identified via culture-dependent and culture-independent techniques.

| Cheese Type, Origin, Quality Label | Milk Type | Technological Characteristics | Microbial Approach | Dominant Bacterial and Fungal Taxa | References |
|---|---|---|---|---|---|
| Bleu d'Auvergne, Auvergne-France, PDO | Cow | LAB, *P. roqueforti* spores Raw/pasteurised milk/dry salt-brined | Culture-dependent | LAB, *Lactobacillus*, *Lactococcus*, *Leuconostoc* | [7,8] |
| | | | Culture-independent (metabarcoding) | *Lactococcus*, *Streptococcus*, *Leuc. mesenteroides*, *Brachybacterium*, *Brevibacterium*, *Lactobacillus*, Enterobacteriaceae, *Romboutsia*, *Acinetobacter* (low proportion) | [8] |
| Cabrales, Asturias-Spain, PDO | Cow or mixtures of cow, sheep, and goat | Autochthonous LAB, *P. roqueforti* spores Raw milk/dry salt | Culture-dependent | *Lc. lactis*, *Lb. plantarum*, *Leuc. mesenteroides*, *Leuc. citreum*, *Lb. paracasei*, *Leuc. pseudomesenteroides*, *Enterococcus durans*, *E. faecium*, *T. koreensis*, *T. halophilus*, *S. equorum*, *Brevibacterium*, *Corynebacterium* *P. roqueforti*, *P. commune*, *P. chrysogenum*, *D. hansenii*, *K. lactis*, *Pich. fermentans*, *Pich. membranaefaciens*, *R. mucilaginosa*, *G. candidum* | [9–12] |
| | | | Culture-independent (PCR-DGGE) | *Lc. lactis*, *Lc. raffinolactis*, *Lc. garvieae*, *Lb. plantarum*, *Lb. casei*, *Lb. kefiri*, *Lb. buchneri* *P. roqueforti*, *P. chrysogenum*, *P. griseofulvum*, *D. hansenii*, *K. lactis*, *C. zeylanoides*, *C. sylvae*, *G. candidum* | [13,14] |
| Danablu, Denmark, PGI | Cow | LAB, *P. roqueforti* spores Pasteurised milk/brined | Culture-dependent | *C. famata*, *C. catenulata*, *C. lipolytica*, *Zygosaccharomyces* spp., *Trichosporon cutaneum* | [15] |
| Gamonedo, Asturias-Spain, PDO | Cow or mixtures of cow, sheep, and goat | No-starters, no-mould spores Raw milk/dry salt/smoked | Culture-dependent | *Lb. plantarum*, *Lb. casei*, *Lb. brevis*, *Lc. lactis*, *Leuc. mesenteroides*, *Leuc. paramesenteroides*, *E. faecalis*, *E. faecium*, *E. durans*, *S. aureus*, *S. epidermidis*, *M. lactis*, *M. varians*, *M. saprophyticus*, *P. roqueforti*, *D. hansenii*, *Cryptococcus laurentii* | [16] |
| Gorgonzola, Lombardy-Piedmont-Italy, PDO | Cow | LAB (*St. thermophilus*, *Lb. delbrueckii*, *Lactococcus* sp.), *P. glaucum*, *P. roqueforti* Raw or pasteurized/dry salt | Culture-dependent | Thermophilic lactobacilli, streptococci, mesophilic lactobacilli, lactococci, micrococci, enterococci, LAB, *Actinomycetota*, *Bacillota*, *Pseudomonadota*, yeasts, moulds (surface) *P. roqueforti*, yeasts | [6,17] |
| | | | Culture-independent (PCR-DGGE) | *S. equorum*, *Brevibacterium linens*, *Corynebacterium flavescens*, *E. faecium*, *Carnobacterium*, *S. saprophyticus* (surface) | [18] |
| Roquefort, Aveyron-France, PDO | Sheep | LAB, *P. roqueforti* spores Raw milk | Culture-dependent | *P. roqueforti*, *Candida*, *Debaryomyces*, *Galactomyces*, *Yarrowia*, *D. hansenii* (*C. famata*), *K. lactis* (*C. sphaerica*), *Candida* spp. (Surface) | [19] |
| Stilton, Nottinghamshire-Leicestershire-Derbyshire-UK, PDO | Cow | LAB (*Lactococcus lactis*), *P. roqueforti* spores Pasteurised milk/dry salt/pierced | Culture-dependent | *Lb. plantarum*, *Lb. brevis* *D. hansenii*, *K. lactis*, *Y. lipolytica*, *Trichosporon ovoides* | [20,21] |
| | | | Culture-independent (PCR-DGGE/TRFLP) | *Lc. lactis*, *E. faecalis*, *Lb. plantarum*, *Lb. curvatus*, *Leuc. mesenteroides*, *S. equorum*, *Staphylococcus* spp. *P. roqueforti*, *D. hansenii*, *K. lactis*, *Y. lipolytica*, *C. catenulata*, *Trichosporon ovoides* | [20,22] |
| Valdeón, León-Spain, PGI | Mixtures of cow and goat | Commercial LAB, *P. roqueforti* spores Raw or pasteurised/dry salt/pierced | Culture-dependent | *Lc. lactis*, *E. faecalis*, *Lb. plantarum*, *Leuc. mesenteroides*, *E. avium*, *E. faecium*, *Lb. casei*, *E. durans*, *Lc. raffinolactis* LAB, Micrococcaceae, Enterobacteriaceae | [23,24] |

Abbreviation key: PDO, protected designation of origin; PGI, protected geographical indication; LAB, lactic acid bacteria; *C.*, *Candida*; *D.*, *Debaryomyces*; *E.*, *Enterococcus*; *G.*, *Geotrichum*; *K.*, *Kluyveromyces*; *Lb.*, lactobacilli; *Lc.*, *Lactococcus*; *Leuc.*, *Leuconostoc*; *M.*, *Micrococcus*; *P.*, *Penicillium*; *Pich.*, *Pichia*; *R.*; *Rhodotorula*; *S.*, *Staphylococcus*; *T.*, *Tetragenococcus*; *Y.*, *Yarrowia*.

### 3.2.1. Microbial Techniques

Culturing methods have been amply used for the characterization of the diversity and succession of the microbial populations throughout cheese manufacturing and ripening before the advent of molecular culture-independent techniques in food microbiology. Among the latter techniques, the temporal temperature gradient electrophoresis (TTGE) and denaturing gradient gel electrophoresis (DGGE) and others have been widely applied to study the spatial and temporal evolution of prokaryotic and eukaryotic communities in several kinds of blue cheese, including Bleu d´Auvergne, Cabrales, Gorgonzola, and Stilton (Table 1). More recently, the application of high-throughput sequencing (HTS) techniques has broadened the bacterial and fungal biotypes detected at both the surface and the interior of many cheese varieties [25,26].

### 3.2.2. Microbial Diversity and Succession in Blue Cheeses

The manufacture of most traditional blue cheeses from raw milk assures a high microbial diversity; even higher if pre-maturation processes are employed [5]. As in many other cheese types, the microbial analysis of blue cheeses has been addressed to the search for and selection of acidifying (LAB) and maturing (*P. roqueforti*) cultures [10,13,15,16,20,24]. As a result of artisan-like manufacture, where uncontrolled environmental conditions are common, large microbial differences between batches, producers, and seasons have been reported [13,14,27].

- Bacterial populations

*Lc. lactis* and *Lc. cremoris*, which reach cell densities up to $10^9$ cfu/g of cheese take over the acidification of the curd; their populations decline slowly during ripening. Other LAB involved include several lactobacilli species such as *Lactiplantibacillus plantarum*, *Lacticaseibacillus paracasei*, and other mesophilic homo- and hetero-fermentative species (e.g., *Levilactobacillus brevis*, *Latilactobacillus curvatus*) develop slowly, although they may surpass the lactococci after 15–30 days of ripening. Dextran-producing *Leuconostoc* (*Leuc. mesenteroides*, *Leuc. citreum*, and *Leuc. pseudomesenteroides*) are frequently found in lower numbers. Among other LAB populations, counts on *Enterococcus*- and streptococci/micrococci-selective media are usually high; these include *E. faecalis*, *E. faecium*, and *Streptococcus* and *Staphylococcus* species (Table 1). More recently, the use of new culturing techniques has allowed for the recovery—as part of the dominant microbiota—of new bacterial species such as *Tetragenococcus* spp., *Staphylococcus equorum*, and species of *Brevibacterium* and *Corynebacterium* genera [unpublished data].

In addition to the bacteria recovered in culture, the DGGE and TTGE techniques allow for the detection of unconventional bacterial types beyond those recovered via culturing, including among other LAB species *Lc. garvieae* and *Lc. Raffinolactis*; *St. thermophilus* [14,27]; and others from the genera *Sphingobacterium*, *Mycetocola*, *Brevundimonas*, etc. [28].

- Yeast and moulds

*P. roqueforti* is the pivotal ripening agent of blue cheeses and is responsible for the visual aspect as well as the texture, taste, and aroma profiles [29]. Nonetheless, particularly in cheeses made from raw milk, a large number of yeasts species can grow accompanying *P. roqueforti* and other fungi; all together, they compose the blue cheese microbiota. Starting from small numbers in milk, yeasts reach majority populations during ripening (up to $10^8$ ufc/g) [30]. Both *P. roqueforti* and yeasts possess potent proteolytic and lipolytic systems that help transform the milk components into flavour compounds. Indeed, selected yeast strains have been proposed as adjuncts and maturing cultures for certain blue cheeses [10,15]. *Geotrichum candidum* (teleomorph state of *Galactomyces candidus*) is among the dominant yeast species in the surface and interior of the cheeses. *G. candidum* produces several enzymes for the breakdown of proteins and fats, resulting in key aroma compounds [31]. Besides *G. candidum*, *Debaryomyces hansenii*, *Kluyveromyces lactis*, *Pichia* spp., *Rhodotorula* spp., *Zygosaccharomyces* spp., and *Saccharomyces* spp. have all been isolated

and identified at different numbers from distinct varieties (Table 1). The microbiota of blue cheeses deserves further characterization, as it may represent a source of new species [32].

Blue-veined cheeses belong to a category of specialty cheeses that are well distinguished from all others by their visual, taste, and aroma profiles. The overall blue cheese quality is thought to result from the concerted action of all members of the microbiota, which, as revealed by the use of state-of-the-art culturing and culture-independent molecular techniques, is formed by an impressive diversity of bacterial and fungal species.

### 3.3. Lactic Cultures

In the production of blue-veined cheeses, natural acidification made by lactic acid bacteria (LAB) has been substituted by the deliberate addition of selected starter cultures. These primary LAB cultures must be able to lower the pH of the milk and survive phage attack. Thus, the main commercially available starter mixtures for blue cheese contain a mixture of strains belonging to the *Lactococcus* genus. Most of the blue cheese types require a mesophilic starter culture, which usually contains strains of *Lc. lactis*, lactis subspecies, and *Lc. cremoris*. These bacteria also contribute to the organoleptic properties of the cheese, generating flavor compounds, either directly by cellular metabolism or indirectly by the release of enzymes. Strains of *Lc. lactis*, subsp. *lactis* biovar *diacetylactis*, are frequently included in mesophilic starter cultures, as this microorganism is able to catabolize citrate to carbon dioxide and the flavor compound diacetyl, which gives the cheese a distinct buttery flavor [33].

Strains of *Leuc. mesenteroides* subsp. *cremoris* are also added due to their ability to produce flavor (diacetyl) but mostly because of its $CO_2$ production, which breaks the structure of the curd, helping the development of the *Penicillium* mould inside the cheese [34].

In those blue cheeses in which the heating of milk and curd is a part of the cheese-making process, a mixture of mesophilic and thermophilic starters can be added. These starters contain mixtures of the strains mentioned and small amounts of *St. thermophilus* and *Lb. delbruecki* subsp. *bulgaricus* [1].

### 3.4. Penicillium roqueforti and Other Adjunct Cultures

*P. roqueforti* is the most important microorganism involved in the manufacture of blue cheese. It belongs to the subgenus *Penicillium* (characterized by ter-/quaterverticillata conidiophorus); colonies grow rapidly (40–70 mm diameter in 7 d, a characteristic feature), plane or lightly radially sulcate, low, and velutinous; conidiophorus walls are very rough [35]. According to Frisvad and Filtenborg [36], there are two varieties, with *P. roqueforti* var. *roqueforti* being the one used in cheese manufacture. In addition, using molecular tools [37] proposed three species that were confirmed later [38]: *P. roqueforti*, *P. carneum*, and *P. paneum*.

*P. roqueforti* appears to have the lowest oxygen requirements for growth of any *Penicillium* [35], which, together with other physiological features such as salt stimulation [39,40] and an ability to grow at low temperatures [41], would explain its presence (even natural) in the interior of blue cheese. On the other hand, this species, like many other *Penicillium* spp., produces several mycotoxins (see Section 3.5).

The unique characteristics of these cheeses are due, to a great degree, on the growth of *P. roqueforti*. The flavour of the final product is mainly due to the lipolytic and proteolytic activities of this fungus [41]. *Penicillium* is the main fungus responsible for the degradation of lipids in this variety of cheese [6,42], with a wide variety of volatile and nonvolatile aroma compounds being produced primarily by *P. roqueforti*. This species is also considered the main origin of the enzymes responsible for proteolysis in blue cheese [42]. In addition, the appearance of the cheese is defined by the blue veins produced by *P. roqueforti* in the interior of the paste. Finally, this species participates in the consumption of lactic acid and neutralization of the cheese [43].

For many years, manufacture of blue cheeses has been carried out in a completely natural way. However, nowadays, the manufacture of these varieties under controlled

conditions and the use of selected *P. roqueforti* strains are common practices in the cheese industry and considered necessary to obtain a product with the desired characteristics. For the selection of the strains several technological properties are evaluated: proteolytic and lipolytic activities, colour, germination and growth rate at the ripening temperatures, salt tolerance, and mycotoxigenicity (see Section 3.5). The proteolytic activity of the strain is extremely important for texture development, whereas the lipolytic ability is essential for aroma development [1]. If proteolysis is not enough, the cheese will be dry and hard, while, if it is in excess, it may be too soft. Additionally, high lipolysis is linked to a more intense flavor (see Section 5). This is considered by the companies offering strains with different properties. Spore suspensions (min. $10^{10}$/mL) of *P. roqueforti* may be added to the milk, to the curd, or during moulding.

Looking at other adjunct starters, yeasts, as it was mentioned in Section 3.2, are part of the natural microbiota of cheeses and may play a role in the manufacture of blue cheeses. Among the list of species found in this variety (more than 20), those that could be used as potential adjunct cultures are *D. hansenii*, *Yarrowia lipolytica*, and *Sac. cerevisiae*, with the first one being the most frequently isolated species in blue cheese [1]. *D. hansenii* has been found in our labs, together with other yeasts such as *Y. lipolytica*, in several Spanish blue cheeses (Valdeón artisanal cheese [44]; Cabrales [10]). As for *Y. lipolytica*, it looks a good candidate to be used in blue cheese manufacture according to its ability to grow and compete with other naturally occurring yeasts such as *D. hansenii* and *S. cerevisiae*, compatibility with and possible stimulation of LAB when co-inoculated and remarkable lipolytic and proteolytic activities [1].

### 3.5. Potential Mycotoxin Production

*P. roqueforti* may produce a range of mycotoxins (toxic secondary metabolites) such as PR-toxin, mycophenolic acid and roquefotines, among others [35,45], and some of them have been found in commercial blue cheeses at very low concentrations. Considering this fact, the relatively low toxicity of the mycotoxins and the instability of some of them (PR-toxin and penicillic acid) means that even a large consumption of blue cheese does not pose a risk to the health of the consumer [1,46–48]. Nevertheless, a selection of strains to be used in cheese manufacture should include a mycotoxin evaluation to ensure the use of those with the lowest mycotoxigenicity.

### 3.6. Pathogens and Spoilage in Blue Cheeses

#### 3.6.1. Pathogenic Microorganisms

In spite of the huge worldwide consumption of cheese, blue cheese and ripened cheeses in general are reasonably safe. Thus, only 152 notifications involving cheese with a risk level that was "serious" or "potentially serious" were found in the RASFF (Rapid Alert System Feed and Food) portal (https://webgate.ec.europa.eu/rasff-window/screen/search, accessed on 1 May 2023); the majority of them (144 notifications) were associated with pathogenic microorganisms, mainly *Listeria monocytogenes*, some serovars of *Salmonella*, and Shiga toxin-producing *Escherichia coli*.

The microbiological criteria laid down in the EU regulations establish the absence of *Salmonella* in 25 g of cheeses (including blue-veined cheeses) from raw milk or milk that has undergone a lower heat treatment than pasteurisation and also the monitoring of *Listeria monocytogenes* under the category of "ready-to-eat foods unable to support the growth of *L. monocytogenes*, other than those intended for infants and for special medical purposes", taking into account the physicochemical properties of blue cheeses [49].

Few outbreaks involving blue cheeses have been reported (Table 2). The first reported incident was associated with Stilton cheese produced in a small dairy cooperative in England. It produced 36 outbreaks of gastrointestinal illness involving 155 cases, and the symptoms were suggestive of staphylococcal food poisoning, but the laboratory testing of cheeses implicated in several of the incidents failed to detect any toxin or chemical, and a single Staphylococcus aureus strain which produced enterotoxin D was isolated from one

suspected sample after enrichment [50]. Another outbreak was caused by norovirus, and the food vehicle was a pasteurised blue cheese dressing, thus being attributed to deficiencies in food handling practices and personnel hygiene [51]. One multistate outbreak was due to contamination with *L. monocytogenes*, affecting 15 patients [52]; mould-ripened cheeses are extremely susceptible to surface contamination during the ripening process [53], and *L. monocytogenes* is regularly associated with cheese rinds in blue cheeses [54,55], having been implicated in a case of listeriosis in Italy [56]. An outbreak due to *Escherichia coli* O157:H7 took place in Scotland in the summer of 2016. The outbreak occurred in two phases and was linked to consumption of a particular type of artisan blue cheese (Dunsyre cheese); the majority of the patients of the first phase of the outbreak declared to have eaten at a hotel where this particular cheese was served. The second phase was linked to a childcare setting probably due to the introduction of the bacteria by an unidentified infected individual with subsequent spread to the childcare group through environmental contamination. A total of 26 confirmed cases were recorded, and 17 of them required hospitalisation. Two cases developed HUS, one of whom, a three-year-old child, died [57].

**Table 2.** Reports of food pathogens associated with blue cheese consumption.

| Agent | Food Vehicle | Year | Illnesses | Hospitalisations/Deaths | References |
|---|---|---|---|---|---|
| Unknown (suspected staphylococcal food poisoning | Stilton cheese (unpasteurised milk) | 1989 | 155 | 1/0 | [50] |
| Norovirus genogroup II | Blue cheese dressing (pasteurised) | 2011 | 3 | 0/0 | [52] |
| *L. monocytogenes* | Blue-veined cheese (unpasteurised milk) | 2011 | 15 | 1/1 | [52] |
| *E. coli* O157:H7 | Dunsyre blue cheese (unpasteurised milk) | 2016 | 26 | 17/1 | [57] |

Other microbiological risks associated with ripened cheeses are the presence of toxic substances produced by microorganisms, such as biogenic amines and mycotoxins.

Biogenic amines can be found in blue cheeses via the proteolysis taking place in the cheese (see Section 4) due to microbial activity [58,59], affecting the quality of the final product. The consumption of food containing higher amounts of toxic biogenic amines may cause food intoxication. In terms of food safety, the most important ones are considered histamine and tyramine. The BIOHAZ Panel of EFSA [60] conducted a qualitative risk assessment of biogenic amines (histamine, tyramine, cadaverine, and putrescine) in fermented foods. The report included ripened cheeses in the main food categories containing biogenic amines.

### 3.6.2. Spoilage

A list of common spoilage agents of blue cheeses is displayed in Table 3. A spoilage microbiota can contaminate and easily grow on the surface of blue cheeses counteracting the activity of microbial starters (*P. roqueforti* and LAB). This fact can cause undesirable changes on cheeses, such as off-flavors or loss of typical colour by other *Penicillium* species. Among them, *P. caseifulvum* has been frequently detected on blue cheeses and cheesemaking facilities and linked to cheese discoloration [61].

**Table 3.** Common spoilage agents of blue cheese.

| Agent | Effect | Cheese Defect | Control |
|---|---|---|---|
| *Pseudomonas* spp. | microbial multiplication | slime; off-flavours | general hygiene; temperature control; proper packaging |
| Lactic acid bacteria | microbial multiplication/production of acid in excess | sourness | general hygiene; temperature control; |
| Yeasts | microbial multiplication | off-flavours; changes in colour (brown) | general hygiene; temperature control; proper packaging |
| Molds | microbial multiplication | changes in colour/flavour | general hygiene; temperature control; ripening control |
| Mites | mites proliferation on the surface | poor appearance | general hygiene; proper clean/disinfection protocols |

It must be noted that *G. candidum* is also reported as both contributing to cheese ripening and sometimes causing a negative interaction by inhibiting *P. roqueforti* used as starter culture [1].

Another biological cause of blue cheese spoilage is the presence of mites from the genus *Tyrophagus*. Mites develop on the surface, probably eating the fungi and causing economic losses as well as health problems (allergies, transmission of microorganisms, and even a prion reservoir) [62].

## 4. Proteolysis and Lipolysis

Proteolysis is the most complex and important primary biochemical process involved in blue cheeses during ripening [63]. It contributes to the softening of the cheese texture via hydrolysis of the protein matrix and a decrease in its aw. In addition, it has a direct effect on flavor through the production of small peptides and amino acids [64].

In blue-veined cheeses, several agents are responsible for extensive proteolysis: proteinases released by starter culture lactic acid bacteria (SLAB) and non-starter lactic acid bacteria (NSLAB); rennet; native milk proteinases; and, especially, proteinases and exo- and endopeptidases produced by *P. roqueforti* [65].

LAB are weakly proteolytic, although they possess a very extensive proteinase/peptidase system with potential to hydrolyze oligopeptides to small peptides and amino acids [66].

Cathepsin D and chymosin produce the glycomacropeptide κ-CN (f106-169) after cleavage of the Phe105-Met106 bond. Likewise, they have similar activities on αs1-casein, but hydrolyze αs2-casein very differently [66]. More important is the proteolytic activity of plasmin in blue cheeses because the pH values established during ripening are close to the optimum for its activity, releasing different peptides [67].

*P. roqueforti* secretes aspartyl and metalloproteinases that have been well characterized, including their specificity on αs1- and β-caseins [66]. In addition, *P. roqueforti* possesses several exopeptidases able to cleave the formed peptides and an extracellular acid carboxypeptidase that releases amino acids. Although the proteolytic activity of *P. roqueforti* varies greatly among strains [68], it is considered the main proteolytic agent in all blue cheeses.

The extensive proteolysis that takes place in blue cheeses is determined by the high % of pH 4.6 soluble nitrogen (pH4.6-SN) ranging from 32.8 to 69.2% [8,24,69]. The slight increase in this fraction at the beginning of ripening is mainly due to the proteolytic activity of chymosin, favored by the low pH and high moisture content of the cheese. Subsequently, after sporulation of *P. roqueforti*, its extracellular proteinases contribute to the rapid increase in pH 4.6-SN. On the other hand, about 77 to 88% of pH 4.6-SN is solubilized in trichloroacetic acid (TCA), showing a deeper proteolysis [24,65,70]. However, there are

exceptions, such as Strachitunt cheese, that presented values lower than 17% pH4.6-SN/TN and 11% TCA-SN/TN [71] associated with a delay in the cheese perforation stage.

At the end of blue cheese ripening, a high degradation of both αs1- and β-casein is reported, with only a small part of them remaining intact [63,65]. The αs1-casein is firstly degraded by the action of rennet on αs1-I-CN, which, as ripening progresses, is a substrate for other enzymes, mainly aspartylprotease from *P. roqueforti* or chymosin [24,72].

During proteolysis, peptides are released that have attracted special interest based on their physiological properties in organisms. These bioactive protein fragments show antimicrobial, antioxidant, antithrombotic, antihypertensive, immunomodulatory, opioid, and antiproliferative activities [73]. Studies of these compounds in blue cheeses are practically lacking, with the exception of Valdeón cheese, which has been previously studied [74]. This study showed the presence of some ACE inhibitory and opioid peptides. Likewise, it was observed that after gastrointestinal simulation, a higher number of bioactive peptides, including antihypertensive, antioxidant, intestinal mucin-secretor, and antibacterial peptides, were found.

The increment in free amino acid (FAA) concentration during ripening is used as an objective index of ripening. Different studies have reported values of 10.11 mg/g in Valdeón cheese [70], 25.01 mg/g in Gorgonzola cheese [65], 47.69 mg/g in Cabrales cheese [75], and 57.32 mg/g in Picón Bejes-Tresviso cheese [76]. The high FAA content has been attributed to the aminopeptidase activity of *P. roqueforti*. Glutamic acid, leucine, valine, lysine, lysine, and phenylalanine predominate in blue-veined cheeses, although tyrosine, serine, and proline are also detected in significant amounts [65,70,72,75,76]. The presence of γ-aminobutyric acid (GABA), a product of the decarboxylation of glutamic acid, has been little studied in blue cheeses. Some studies Redruello et al. [77] have reported concentrations between 1000 and 4000 mg GABA/kg in Cabrales, Gamonedo, and Picón Bejes-Tresviso cheeses, being much higher than those described for other types of bacteria-ripened cheeses. This compound has gained great relevance in recent years as it presents bioactive properties with beneficial health effects [78].

Due to the extensive and profound proteolysis that takes place in blue cheeses, the level of biogenic amines is higher than in other varieties without moulds. The major biogenic amines in blue-vein cheeses are tyramine, cadaverine, putrescine, and histamine [70,79].

Lipids in cheese can undergo hydrolytic or oxidative degradation. Oxidative changes are very limited due to the low oxidation–reduction potential (around −250 mV). In cheeses, hydrolysis of triglycerides by lipases with release of fatty acids (FFAs) during ripening is more important [64]. At the end of ripening, blue cheeses show a very high FFA concentration as a result of strong lipolysis, being variable in function depending on the cheese type: Picón Bejes-Tresviso with 58,355 mg/kg [76], Gamonedo with 75,685 mg/kg [16], Bleu d'Auvergne and Fourme D'Ambert with 86,000 and 30,000 mg/kg, respectively [80], and Valdeón PGI cheese with 42,500 mg/kg [81].

Major FFAs are oleic (C18:1), palmitic (C16:0), and myristic (C14:0) acids [76,80]. In some cheeses, a decrease in FFA concentration is observed at the end of ripening, attributed to their degradation via the oxidative pathway [82]. The cheeses produced with *P. roqueforti* strains had a higher abundance of volatile compounds such as methyl ketones and secondary alcohols [29].

Ketones are the main aroma compounds in blue cheeses, which represent 50–75% of the total aroma profile in Roquefort, Bleu des Causses, and Bleu d'Auvergne [83]; 47–55% in Gorgonzola [84]; and 55–75% in Stilton [85]. Primary and secondary alcohols are, after ketones, the most important compounds in the aroma of blue cheeses, representing more than 30% of the volatile compounds in Gorgonzola [84], from 10–30% in Stilton [85], and from 15–20% in Roquefort [83]. Alcohol can be formed by enzymatic reduction of methyl ketones using *Penicillium* spp. [82]. At the end of ripening, 3-methyl butanol is the predominant alcohol in blue cheeses, although high concentrations of 2-pentanol, 2-heptanol, and 2-nonanol have also been detected [86], responsible for the characteristic aroma of blue cheeses. Finally, there are esters that contribute to attenuating the pungent flavor

typical of methyl ketones [84]. Ethyl esters together with methyl esters are the predominant compounds [87], with ethyl butanoate and ethyl hexanoate being the most prominent.

## 5. Sensory Characteristics

The colour of the inner part is white-light yellow (depending on the type of milk used) with more or less regularly distributed blue-green mold veins caused by *P. roqueforti* (the colour is dependent on the strain used). Openings of piercing channels may be visible. As was mentioned previously, an open texture, with a minimum amount of oxygen, is necessary to allow for the growth of *P. roqueforti*. Other sensory features are a consequence of the intense proteolysis and lipolysis taking place inside, as indicated before. The texture is more or less soft, smooth, and creamy. In some types, it may be sliceable and spreadable, or it may crumble when cut. The smell is usually intense, pleasant, and penetrating. The characteristic odour impressions originate from the methyl ketones, introducing fruity, floral, and spicy notes. In the smoked varieties (Gamonedo, Spain), the aroma is a little smoky. As for the taste, it is usually intense and sharp, relatively spicy, salty, and acidic. The rind used is natural, soft, thin, creamy and with different colours (orangey-brown, greyish, reddish, or yellow) caused by microbial growth on the surface. In Danablu it is white and free of bacterial or mould growth. In some varieties (Cabrales and Roquefort), the cheese is wrapped up with aluminum foil when it is ready for consumption [3,4].

## 6. Conclusions

The sensory characteristics of blue cheeses and, ultimately, the essence of these varieties are based on complex biochemical reactions due, to a large extent, to a diverse microbiota in which fungi and bacteria participate in an active way. Although there have been studies on the microbiology and biochemistry of blue cheeses in recent years, further research is needed, in particular in the characterization of artisanal cheeses. This would allow us to keep the global diversity of existing blue cheeses which enriches the broad list of cheese varieties available to the consumer. In addition, more research is needed to elucidate the role of bioactive compounds generated during ripening, such as GABA or bioactive peptides, on the functionality of these varieties.

**Funding:** Research in this area has been supported by projects from the Spanish Ministry of Science and Innovation (PID2019-110549RB-I00/AEI/10.13039/501100011033) and Asturias Principality (AYUD/2021/50916; AYUD/2021/57336).

**Informed Consent Statement:** Not applicable.

**Data Availability Statement:** No new data were created or analyzed in this study. Data sharing is not applicable to this article.

**Conflicts of Interest:** The authors declare no conflict of interest.

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
