# Peer review of "Blue Cheeses: Microbiology and Its Role in the Sensory Characteristics"

_2624-862X, doi:10.3390/dairy4030027_

Round 1

Reviewer 1 Report

The manuscript is well written and fully brings an interesting theme to an area.

I only have small suggestions for this manuscript:

The title needs to be better developed

Standardize the abbreviations of the scientific names in the Tables and throughout the manuscript

Several paragraphs lack references

Double check the use of italics in the text (example topic 3.4)

Author Response

The title needs to be better developed

  • See the new title:

Blue cheeses: microbiology and its role in the sensory characteristics

Standardize the abbreviations of the scientific names in the Tables and throughout the manuscript

  • Revised, see lines 127, 182, 86, 306. In addition, an error was corrected in Table 1 (line 15 of the Table).

Several paragraphs lack references

  • We just found one paragraph with may have few references. See amendment in line 227.

Double check the use of italics in the text (example topic 3.4)

  • The name Penicillium roqueforti is not in italics to differentiate it from the rest of the title, that is in italics. The rest of the text has been reviewed.

Reviewer 2 Report

Dear Authors,

the manuscript deals with an interesting review on the Blue cheeses. The paper is well written. It considers the sensory characteristics of blue cheeses and its microbiota. It would be desirable to improve research on these cheeses and increase the information available to the consumer.

Best regards

Author Response

Thank you for the comments.

Reviewer 3 Report

Overall this submission is well written, is clear and covers many aspects of blue cheese manufacture and ripening.    While it does not enter into the detail of taxonomy of the microbiota of blue cheeses, it provides sufficient detail on the topic.   Some further detail on the changes to pH, texture and microstructure of blue cheese during ripening would be welcome as well as on the role of calcium levels in the curd/cheese after manufacture.  The submission is very focused on microbiology but requires some further detail on the technology and chemistry of blue cheese ripening.  The submission finishes very abruptly and a conclusions paragraph is suggested particularly focusing on recommendations for future research.  Similarly, there is a disproportionate number of older references in the bibliography.

Author Response

Some further detail on the changes to pH, texture and microstructure of blue cheese during ripening would be welcome as well as on the role of calcium levels in the curd/cheese after manufacture.  The submission is very focused on microbiology but requires some further detail on the technology and chemistry of blue cheese ripening.

  • The manuscript is focused mainly on microbiology and the biochemistry of ripening, although it also includes general information about them (types and processing main steps), and sensory characteristics. Nevertheless, some lines were added to section 3.1. See also the new title which focus the topics mainly covered in the manuscript.

The submission finishes very abruptly and a conclusions paragraph is suggested particularly focusing on recommendations for future research. 

  • A final Conclusions section has been added.

Similarly, there is a disproportionate number of older references in the bibliography.

  • This is a review, so old references may be necessary.